# ALIGNING LARGE LANGUAGE MODELS WITH DOMAIN ADAPTATION

## ABSTRACT

Aligning large language models (LLMs) has emerged as a critical challenge in the age of generative AI: LLMs must be appropriately aligned with human values and preferences in order to be helpful and harmless. In many real world cases, however, large amounts of preference data are not available on important tasks, limiting the effectiveness of resulting reward models. In some cases, data from a similar task is available, and unlabeled data on the target task is available or can be generated by an LLM. In other cases, clean data may be available to train an LLM for real-world use on noisy data, small amounts of labeled data on the target task may be available, or data may be available on an easier task. In this work, we demonstrate that domain adaptation can effectively use different types of data, by transferring supervision and human values across tasks with similar data distributions, strengthening resistance to noisy data, improving few-shot generalization ability, and even transfer from easy to hard tasks, in the form of short to long generalization. Specifically, we propose Data Efficient Alignment for Language (DEAL), using domain adaptation to effectively perform cross-task alignment in scenarios where labeled target data is not available. We evaluate our method for reward model training on a variety of benchmarks and demonstrate that our method can meaningfully improve performance on target tasks by utilizing data on related tasks or low amounts of data. Furthermore, we offer analysis on the inner mechanism of domain adaptation and the alignment of embedding distributions.

## 1 INTRODUCTION

Large language models (LLMs) have become increasingly powerful, reaching or surpassing human abilities on a variety of tasks. On real-world applications, LLMs are especially successful when large amounts of labeled data are available to perform fine-tuning, either through reinforcement learning from human feedback (RLHF) with preference data or supervised fine-tuning (SFT) with ground truth completion data.

However, on many real-world tasks, large amounts of preference data on target tasks are not available, as data labeling may be very expensive. For example, in low-resource languages, low amounts of supervised data may be available, especially in specialized tasks such as those relating to law or medicine. Furthermore, large amounts of one to one translation data may not be available from a high-resource language to the given low resource language. In these cases, however, large amounts of unlabeled data in the target language may be available, though it may not be matched to the high-resource labeled data. Despite the lack of target data, the labeled high-resource data still provides highly relevant supervision that can benefit performance on the low-resource language task. Still, direct training on the high-resource data may not fully transfer relevant supervision to the target task. Instead, domain adaptation techniques allow for explicit distribution matching between the source and target data by leveraging knowledge about the unlabeled target data distribution and aligning the two data distributions. This allows common skills and features to be transferred between the two tasks and ultimately improves performance on the target task without using any labeled target data.

On other tasks, while labels on clean, labeler or LLM generated data may be available, labels may not be available for the target distribution of real-world, noisy data (e.g. user-generated internet data). In addition, such noise reduction may be useful for LLM post-training (e.g. RLHF, DPO, rejection sampling), in which the distribution of LLM generated outputs may shift away from the

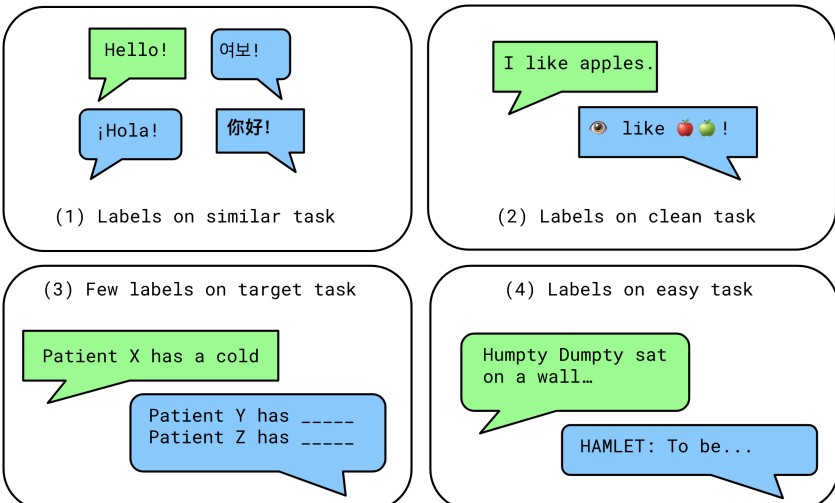

Figure 1: Description of 4 scenarios where DEAL improves LLM alignment, in the absence of large amounts of target data. From top left clockwise: **(1)** Labeled data is available on a similar task (e.g. English), along with unlabeled data on the target task (e.g. Korean). **(2)** Labels are available on a "clean" set of data (e.g. "I like apples.") and unlabeled real-world, noisy data is available (e.g. with emojis, slang). **(3)** Very few labels are available on the target task, along with large amounts of unlabeled data. **(4)** Labels are available on an easy task (e.g. nursery rhymes), along with unlabeled data on a hard task (e.g. Shakespeare) on which labels are expensive.

distribution of training preference data points. In these cases, simply training on the clean data may not give desired performance on the real world distribution of data, which may lie slightly outside of the training data distribution. Domain adaptation aligns unlabeled noisy samples to samples in the training dataset with labels, allowing for noise resistance and transfer of supervision from the clean, narrower distribution to the noisy, broader distribution.

On other tasks, while data labeling may be expensive (e.g. specialists are required), there may be small quantities of labeled data available, though large amounts of unlabeled data may be available (or can be generated). In these cases, direct training on the small amounts of labeled data may not give the desired performance. Knowledge of the target data distribution can greatly improve performance by informing the model of how the target unlabeled distribution relates to the given labeled data points, allowing it to perform principled generalization from the labeled data points to the larger, unlabeled data distribution. Domain adaptation, with explicit distribution alignment between the source (labeled) and target (unlabeled) distributions, allows the model to match unlabeled data points to similar labeled data points. This ensures that the model maximally utilizes the small amounts of labeled data and draws a decision boundary that separates all of the unlabeled data.

In other cases, labeled data may only be available for easier subtasks that are part of the larger target task. For example, while labeling long books and articles may be prohibitively expensive, labeling individual sentences or paragraph fragments is manageable and less expensive. While some skills may be unique to the longer texts (e.g. long-term plot development), many of the skills required to distinguish short and long texts may be similar, including grammar, punctuation, and logic. Through domain adaptation, supervision can be effectively transferred from the easier to label short task to the label-expensive long task, by explicitly aligning these commonalities between the two tasks. This reduces the need for costly labels on difficult tasks, rather allowing humans to express their preferences on easier tasks and transferring these preferences to other tasks.

All in all, we identify four real-world LLM alignment scenarios with a lack of target data, including transferring supervision across similar tasks (e.g. between languages), transferring from clean to noisy data (e.g. internet text), using only a few target examples (labels are expensive), and transferring human preferences from easy tasks to hard tasks. A diagram depicting these four tasks is given in Figure 1.

We propose **D**ata **E**fficient **A**lignment for **L**anguage (DEAL), using domain adaptation to learn domain-invariant representations for better generalization. DEAL effectively aligns capabilities and values of LLMs across related tasks, thereby improving performance on the target task using little to no labeled data on the target task.

In summary, our contributions are as follows:

1. We propose DEAL (Data Efficient Alignment for Language), using domain adaptation to transfer supervision and skills across tasks, improve LLM abilities on target tasks, and offer analysis as to the reasons behind its effectiveness.

2. We demonstrate that DEAL can successfully improve performance in cross-lingual transfer to low-resource languages on a Reddit preference task.

3. We show that DEAL improves resistance to noise and transfers supervision from a set of formal, clean data to noisy, real-world data on the Reddit preference task.

4. We demonstrate that DEAL meaningfully improves few-shot generalization ability using small amounts of data on a safety task and effectively combines few-shot examples with unlabeled data on the target task to improve performance.

5. We show that DEAL can identify commonalities between data on smaller, easier, tasks and larger, harder tasks with expensive labels. DEAL improves performance on the target task by transferring these common capabilities from the easy task to the hard task.

## 2 RELATED WORK

**Cross-task alignment of LLMs**: The problem of alignment, or ensuring that LLMs properly adhere to human values, has become more important as LLMs increase in ability. Techniques such as Reinforcement Learning from Human Feedback (Ouyang et al., 2022) have been used to fine-tune LLMs to match human preference data, and have been successfully applied to commercial, user-facing LLMs. Recently, attempts have been made to measure the generalization ability of aligned LLMs by varying the training and evaluation tasks. For example, Hase et al. (2024) found that LLMs trained on easy STEM and general-knowledge questions showed a surprisingly high zero-shot generalization ability to harder questions. Sun et al. (2024) found that reward models exhibited a stronger generalization ability than LLMs trained using supervised fine-tuning (SFT) when transferring from easy to hard math questions. For easy-to-hard generalization, Zhou et al. (2022) used least-to-most prompting to break up a hard task into easier tasks, allowing for the direct use of LLMs trained on easier tasks on hard tasks. Furthermore, Wu et al. (2024) and Li et al. (2024) have shown that training an LLM on a source language shows significant zero-shot generalization to other languages.

**Domain Adaptation**: Domain adaptation, or the problem of transferring supervision from a source task with a large amount of labeled data to a target task with little to no labeled data (but potentially large amounts of unlabeled data), has found many promising real-world applications, such as self-driving (Li et al., 2023), where the problem is to transfer a model (e.g. for object detection) trained on images in one condition (e.g. sunny) to another (e.g. rainy) and Sim2Real transfer for robotics (Truong et al., 2020), where the problem is to effectively transfer an agent trained on simulation data to the real world. Representative approaches to domain adaptation include Maximum Mean Discrepancy (Tzeng et al., 2014), which optimizes source classification error while maximizing domain confusion in the hopes of creating domain-invariant feature representations. Another representative approach for domain adaptation is Domain Adversarial Neural Networks (DANN) (Ganin et al., 2015), which uses a reverse gradient from a discriminator head to minimize the difference between source and target feature representations while successfully completing the source task, with the hope that this distributional alignment of feature representations will effectively transfer supervision from the source to target task. A separate class of approaches involves learning a data mapper between the source and target data distributions that maps a source example to a target example and vice versa, for example CycleGAN (Zhu et al., 2017). In addition, other extensions to DANN have been proposed, such as Deep Joint Distribution Optimal Transport (DeepJDot) (Damodaran et al., 2018), which aligns the joint distribution of feature representations and labels and Wasserstein Distance Guided Representation Learning (WDGRL) (Shen et al., 2018), which attempts to address the instability of domain adversarial training by applying ideas from Wasserstein GANs (Arjovsky et al., 2017) to domain adaptation.

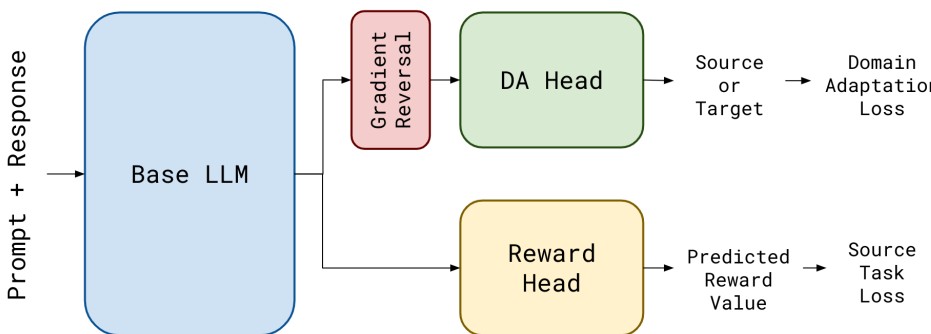

Figure 2: An illustration of DEAL, using domain adaptation for aligning large language models.

## 3 METHOD

While prior approaches to cross-task LLM alignment have primarily focused on measuring the zero-shot generalization ability of LLMs trained on a source task, we study techniques for explicitly improving this generalization ability and ensuring awareness of the target evaluation data distribution. We propose the application of domain adaptation to LLMs in order to align embedding distributions between two separate tasks, with labeled and unlabeled data. Our method, DEAL, allows for the generalization to be tailored to the specified target distribution, and allows the reward model to learn the transformation between the two distributions, rather than relying on guesses as to the underlying transformation.

We focus on training reward models with a randomly initialized classification head on top of the final embedding of the LLM for two reasons:

1. Prior work (Sun et al., 2024) has shown that reward model training allows for better generalization ability compared to supervised fine-tuning.
2. Our trained reward models can then be used with RLHF and best-of-N sampling to effectively align downstream LLMs with the trained reward models.

Specifically, we apply Wasserstein Distance Guided Representation Learning (Arjovsky et al., 2017) (WDGRL), a more stable domain adaptation algorithm, on the final embedding from the LLM before the classification head. During training time, each sampled batch contains some source labeled data and some target unlabeled data. Embeddings are calculated for both the source and target data, and predicted rewards are calculated for the labeled source examples. We then calculate the WDGRL loss and main source task loss.

More formally, given $n_s$ source and $n_t$ target examples, source and target embeddings $e_s, e_t$, and source labels $y_s$, we learn a main task head $f_{\text{main}}$ and a domain critic head $f_{\text{critic}}$. The critic head outputs scores that attempt to discriminate between source and target embeddings, as given by the Wasserstein distance loss, and is regularized with a gradient penalty. We offer an illustration of our method in Figure 2.

The domain critic attempts to maximize the WDGRL loss:

$$e_{\text{mix}} = \text{sample random linear combinations between } e_s \text{ and } e_t \quad (1)$$

$$e = \text{concat}(e_s, e_t, e_{\text{mix}}) \quad (2)$$

$$\mathcal{L}_{\text{GP}} = (\|\nabla_e f_{\text{critic}}(e)\|_2 - 1)^2 \quad (3)$$

$$\mathcal{L}_{\text{WD}} = \frac{1}{n_s} \sum_{i=1}^{n_s} f_{\text{critic}}(e_{s,i}) - \frac{1}{n_t} \sum_{i=1}^{n_t} f_{\text{critic}}(e_{t,i}) \quad (4)$$

$$\mathcal{L}_{\text{WDGRL}} = \mathcal{L}_{\text{GP}} - \gamma \mathcal{L}_{\text{WD}} \quad (5)$$

Intuitively, by maximizing the WD loss, the critic is attempting to increase predicted critic scores for the source examples and decrease those for the target examples. In addition, a gradient penalty

loss is employed to ensure that the critic's function surface is smooth and optimizable by the feature extractor. A gradient reversal layer is placed in between the feature extractor and critic head, so that the objective of the base LLM is to create embeddings that minimize the WDGRL loss and are hence indistinguishable between source and target.

In total, the feature extractor attempts to minimize a combination of the WDGRL loss on both source and target examples and the main task loss on source examples:

$$\mathcal{L}_{\text{total}} = \mathcal{L}_{\text{WDGRL}} + \lambda\mathcal{L}_{\text{task}} \tag{6}$$

Through this adversarial training, the critic attempts to pick up on features in the embeddings to discriminate between source and target examples, while the feature extractor attempts to eliminate these differences. The feature extractor is incentivized to learn representations that are indistinguishable between source and target domains and informative to do well on the source task, thereby aligning the embedding distributions and transferring supervision from the source task to the target task.

## 4 TOY EXPERIMENT: ODD ONE OUT

To gain more insight into the inner mechanisms of DEAL and distribution alignment, we studied a toy task of identifying the object that is the odd one out from a list of 5 objects. Example:

> Apple, Banana, Grape, Pencil, Cherry → odd one out is **Pencil**

We generated sets of 100 concepts belonging to 5 categories of foods: desserts, fruits, sauces, vegetables, and snacks. We then created one set of data for each category, where each example consisted of 4 items from that category and one item not from that category, where the goal of the LLM was to identify the item that did not fit. Examples of concepts are given below:

> Desserts: Cake, Pie, Ice Cream, Cookies, Brownies
> Fruits: Apple, Banana, Orange, Grape Strawberry

Table 1: Accuracy results for DEAL on odd one out. Results over 3 seeds ($\bar{x} \pm s_{\bar{x}}$). Random = 0.2

| Source | Target | Train on source | DEAL |
|--------|--------|-----------------|------|
| Desserts | Fruits | $0.363 \pm 0.090$ | $\mathbf{0.686 \pm 0.035}$ |
| Desserts | Sauces | $0.400 \pm 0.107$ | $\mathbf{0.743 \pm 0.028}$ |
| Desserts | Vegetables | $0.234 \pm 0.057$ | $\mathbf{0.489 \pm 0.037}$ |
| Desserts | Snacks | $0.477 \pm 0.045$ | $\mathbf{0.728 \pm 0.013}$ |
| Fruits | Desserts | $0.261 \pm 0.060$ | $\mathbf{0.561 \pm 0.047}$ |
| Fruits | Sauces | $0.254 \pm 0.101$ | $\mathbf{0.404 \pm 0.019}$ |
| Fruits | Vegetables | $0.649 \pm 0.016$ | $\mathbf{0.714 \pm 0.002}$ |
| Fruits | Snacks | $0.091 \pm 0.017$ | $\mathbf{0.455 \pm 0.038}$ |
| Sauces | Desserts | $0.398 \pm 0.035$ | $\mathbf{0.610 \pm 0.018}$ |
| Sauces | Fruits | $0.251 \pm 0.008$ | $\mathbf{0.442 \pm 0.037}$ |
| Sauces | Vegetables | $0.285 \pm 0.056$ | $\mathbf{0.454 \pm 0.037}$ |
| Sauces | Snacks | $0.405 \pm 0.071$ | $\mathbf{0.643 \pm 0.005}$ |
| Vegetables | Desserts | $0.174 \pm 0.029$ | $\mathbf{0.389 \pm 0.019}$ |
| Vegetables | Fruits | $0.708 \pm 0.012$ | $\mathbf{0.756 \pm 0.010}$ |
| Vegetables | Sauces | $0.202 \pm 0.041$ | $\mathbf{0.538 \pm 0.061}$ |
| Vegetables | Snacks | $0.247 \pm 0.092$ | $\mathbf{0.471 \pm 0.040}$ |
| Snacks | Desserts | $0.304 \pm 0.018$ | $\mathbf{0.629 \pm 0.033}$ |
| Snacks | Fruits | $0.183 \pm 0.012$ | $\mathbf{0.397 \pm 0.027}$ |
| Snacks | Sauces | $0.248 \pm 0.029$ | $\mathbf{0.683 \pm 0.068}$ |
| Snacks | Vegetables | $0.229 \pm 0.016$ | $\mathbf{0.396 \pm 0.064}$ |
| Average | | $0.318 \pm 0.022$ | $\mathbf{0.559 \pm 0.018}$ |

While a naive LLM trained on one category of data may learn the shortcut of simply outputting lower reward for any choice that does not belong to that specific category (i.e. Output the one that isn't a fruit), we conducted an experiment to measure the ability of the LLM to understand the general task of identifying the item that does not fit (i.e. Output the item that is different from the other 4), by training and evaluating on two different categories of data. Results are given in Table 1.

Our results show that applying domain adaptation between domains allowed the LLM to transfer supervision from one task to the other, despite not having labels on the target task. Instead of simply selecting the item not belonging to a particular category, DEAL forces the LLM to identify the general pattern between the source and target tasks and improving performance on the target task.

## 5 EXPERIMENTS

We provide a overview of our experiments testing each of the four scenarios in Figure 1. For all experiments, we select the snapshot with maximum validation performance and evaluate on a held-out test set. Detailed experimental settings are given in Appendix A.

### 5.1 SCENARIO 1: TRANSLATION

Table 2: Accuracy results on the translation task. Results are over 3 seeds, $\bar{x} \pm s_{\bar{x}}$. Random = 0.5

| Split | Language | Train on source | DEAL |
|---|---|---|---|
| legaladvice | Korean | $0.595 \pm 0.005$ | $\mathbf{0.679 \pm 0.005}$ |
| | Thai | $0.627 \pm 0.017$ | $\mathbf{0.656 \pm 0.006}$ |
| | Chinese | $0.613 \pm 0.015$ | $\mathbf{0.678 \pm 0.032}$ |
| | Average | $0.611 \pm 0.008$ | $\mathbf{0.671 \pm 0.010}$ |
| askscience | Korean | $0.572 \pm 0.015$ | $\mathbf{0.634 \pm 0.003}$ |
| | Thai | $0.618 \pm 0.018$ | $\mathbf{0.681 \pm 0.002}$ |
| | Chinese | $0.594 \pm 0.007$ | $\mathbf{0.624 \pm 0.008}$ |
| | Average | $0.594 \pm 0.010$ | $\mathbf{0.646 \pm 0.009}$ |
| explainlikeimfive | Korean | $0.651 \pm 0.013$ | $\mathbf{0.678 \pm 0.011}$ |
| | Thai | $0.633 \pm 0.003$ | $\mathbf{0.653 \pm 0.004}$ |
| | Chinese | $\mathbf{0.684 \pm 0.006}$ | $0.667 \pm 0.006$ |
| | Average | $0.656 \pm 0.008$ | $\mathbf{0.666 \pm 0.005}$ |

To evaluate the ability of our method to effectively perform alignment when given unlabeled data on a similar task, we applied DEAL to a translation task on the Stanford Human Preferences (Etha-yarajh et al., 2022) (SHP) dataset. The SHP dataset consists of questions from Reddit and pairs of preferred and non-preferred answers. We selected a diverse set of three splits (legaladvice, askscience, explainlikeimfive), and translated the prompts and responses to a diverse set of three languages (Korean, Thai, Chinese) using NLLB (team et al., 2022). Example:

> askscience English: If the universe is expanding in all directions how is it possible that the Andromeda Galaxy and the Milky Way will collide?
>
> askscience Chinese translated: 既然宇宙在各个方向扩大, 那么安德罗米达星系和银河系怎么可能会碰撞?

We then evaluated both a train on source baseline (training on the English split) and DEAL. Results are given in Table 2. Our results indicate that domain adaptation is able to consistently improve performance at ranking Reddit responses across languages, especially in specialized domains such as law and science, highlighting the potential of our method to improve cross-lingual transfer capabilities of modern LLMs and ensure that they provide effective responses in low-resource languages. We note that in the single case where domain adaptation slightly decreases performance, we believe this is due to a high zero-shot performance of training on source, leaving little to no room for improvement in transfer ability using domain adaptation.

Table 3: Accuracy results on the noise task. Results are over 3 seeds, $\bar{x} \pm s_{\bar{x}}$. Random = 0.5

| Split | Train on source | DEAL | Train on target |
|---|---|---|---|
| legaladvice | $0.706 \pm 0.001$ | $\mathbf{0.755 \pm 0.014}$ | $0.767 \pm 0.003$ |
| askscience | $0.634 \pm 0.015$ | $\mathbf{0.649 \pm 0.013}$ | $0.674 \pm 0.014$ |
| explainlikeimfive | $0.674 \pm 0.005$ | $\mathbf{0.704 \pm 0.012}$ | $0.740 \pm 0.002$ |
| Average | $0.671 \pm 0.011$ | $\mathbf{0.703 \pm 0.017}$ | $0.727 \pm 0.014$ |

Through this experiment, we demonstrate that DEAL is able to effectively align LLMs with human preferences on a target task even without large amounts of target labeled data, instead relying on a set of labeled data in a different language and performing cross-language generalization. We believe that our work on cross-language alignment is an important step forward towards building an all-language LLM, where concepts in different languages are aligned, so that fine-tuning in one language automatically transfers to all other languages and democratizing knowledge for all languages.

## 5.2 Scenario 2: Noise Generalization

Next, we examine the scenario where clean, professional or LLM-generated data is available, but we desire that the reward model perform well on noisy, real-world data, such as that on the internet. We selected the three splits of the Stanford Human Preference dataset that we used for the translation task and used Gemma-2-9b-it (Riviere et al., 2024) to rewrite questions and responses with formal language. Example:

> Question: Explain like I'm five years old: Why can your body have a "sleep debt" but not a "sleep surplus"? Why does my 15 hours of sleep on the weekend not counteract the 4 hours I get on a weeknight?
>
> Original (informal): Its like your laptop - once you use is for a while you need to charge it up and if it falls to 0 you need to charge it before it works again but if you leave it overnight the max is only 100%
>
> Formal: It is analogous to a laptop computer; once it has been utilized for a period of time, it requires recharging. If the charge depletes to zero, it must be recharged before it can function again. However, if it is left unattended overnight, the maximum charge attainable is 100%.

As illustrated in this example, the real world data contains a significant amount of noise in the form of spelling, grammar, and punctuation errors, as well as abbreviations such as (max). A reward model trained on only the formal data may not transfer well to the internet data, due to the "noisy" language. We evaluate the ability of DEAL to improve the reward model's generalization to this noisy data. Results are given in Table 3.

We demonstrate that DEAL is able to effectively transfer supervision to harder, noised, real-world data even when only given labels on a set of formal, "easy" data. Furthermore, we believe that DEAL has the potential to benefit applications in the problem of superalignment (Burns et al., 2023), where labels from a potentially noisy human labeler are used to supervise a superhuman LLM, as DEAL can effectively align supervision from the human to existing knowledge in the LLM and eliminate the noise differences between the two. DEAL is able to effectively strengthen the generalization ability of the reward model and improve its resistance to noise, allowing for more effective reward models to be used in the real world.

## 5.3 Scenario 3: Few-shot Generalization

Next, we examined the scenario where a small amount of target data is available, along with large amounts of unlabeled data. For this experiment, we used an English translation of the CValues dataset (Xu et al., 2023).

We sampled 10 examples from the training dataset and using these as the labeled training data. We then used the entire training dataset as the unlabeled data and attempted to use DEAL to transfer

Table 4: Accuracy results for DEAL for few-shot transfer ($\bar{x} \pm s_{\bar{x}}$ over 3 seeds). Random = 0.5

| Method | Split A | Split B | Split C | Average |
|---|---|---|---|---|
| Train on few only | $0.820 \pm 0.011$ | $0.862 \pm 0.005$ | $0.852 \pm 0.021$ | $0.845 \pm 0.009$ |
| DEAL | $\mathbf{0.852 \pm 0.042}$ | $\mathbf{0.965 \pm 0.003}$ | $\mathbf{0.943 \pm 0.005}$ | $\mathbf{0.920 \pm 0.021}$ |
| Train on all data | - | - | - | $0.999 \pm 0.000$ |

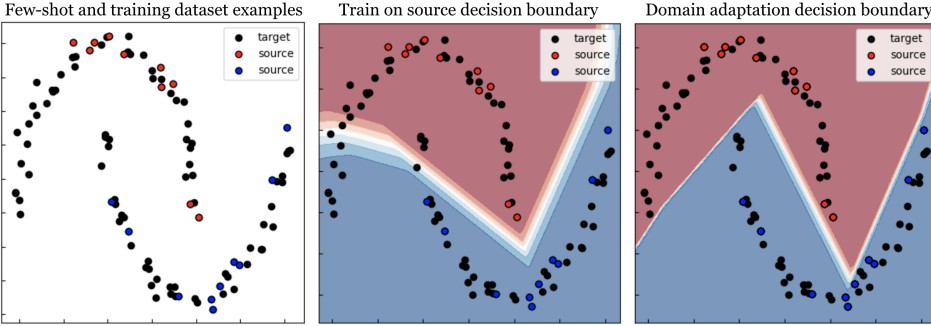

Figure 3: Decision boundaries for zero-shot training on source and domain adaptation on a toy two moons dataset. Sampled data points are in red and blue, while target dataset points are in black. Each moon is a separate class. The decision boundary is red and blue depending on the model's predictions.

supervision from the labeled data to the unlabeled data. Intuitively, distribution alignment clusters the target data around the few-shot examples, improving generalization ability. Performance is given in Table 4.

Our results indicate that DEAL is able to significantly improve performance on the target dataset using only unlabeled target data and 10 few-shot examples. We believe that the reason why DEAL is successful for few-shot generalization is that having knowledge of both a few labeled examples *and* the distribution of unlabeled data (which provides much better information about the evaluation distribution) allows the model to align the unlabeled data with known labeled data (essentially clustering the unlabeled data around the labeled data), and creating a decision boundary that more effectively separates both the labeled data and the unlabeled data.

To verify this hypothesis, we ran a toy 2D experiment on the classic two moons dataset, performing few-shot generalization using domain adaptation. We sampled a set of few-shot data points which only covered a portion of the two moons, and observed that domain adaptation was able to successfully improve performance, as shown in Figure 3. Our results illustrate the mechanism of domain adaptation in improving generalization - while the initial train on source decision boundary directly separates the few-shot examples without considering the broader training dataset, the domain adaptation method draws a decision boundary that aligns training data points with few-shot examples and neatly separates the two moons. Furthermore, on the safety task, we bulk evaluated and saved embeddings for both the train on source baseline (after training 10 epochs) and the domain adaptation method (after training 1 epoch) on both the set of 10 few-shot examples and 1000 randomly selected examples from the larger training dataset. We then learned a PCA transformation on the few-shot embeddings and applied this transformation to the 1000 embeddings. We plotted both positive and negative examples, as shown in Figure 4, which shows PCA reduced embeddings as points for seed 0 of split A.

The few-shot examples in the train on source baseline appear on the edges of the main mass of training dataset points, indicating that the representations for the main training dataset may be farther away from the few-shot examples. On the other hand, in the domain adaptation plot, the main training dataset examples are solidly clustered around the few-shot examples, indicating that DEAL

has successfully aligned the few-shot and training distributions, and provides insight into the reason behind increase in performance. This trend of matching training points and few-shot points is consistent throughout different splits and seeds. While previous work on aligning LLMs with few examples has focused on few-shot prompting, we demonstrate the potential for *few-shot fine-tuning* of LLMs. In other words, we show that LLMs can be aligned on a new task given very few examples and large amounts of unlabeled data, which can even be generated by an LLM. Our results illustrate the potential of DEAL for drastically reducing the amount of labels necessary to effectively align an LLM on a task.

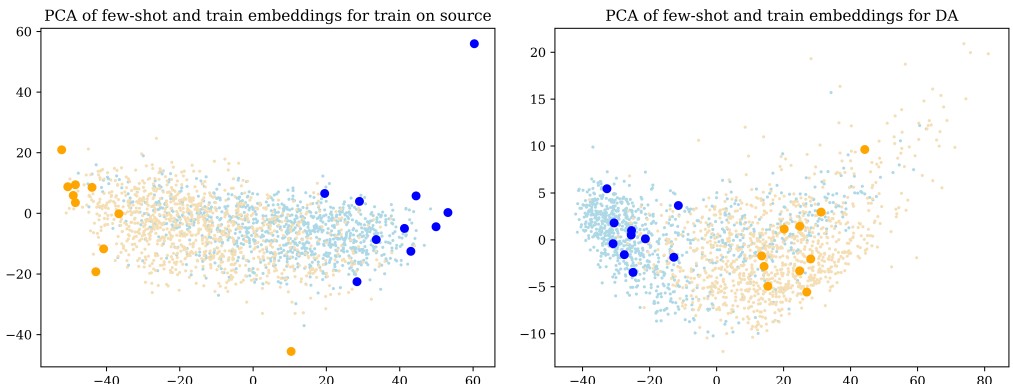

Figure 4: PCA reduced points of embeddings for few-shot examples and 1000 sampled examples from the training dataset. Few-shot examples are large, training points are smaller. Positive examples are blue, negative examples are orange. Left: Train on source baseline, Right: DEAL.

## 5.4 Scenario 4: Short-to-Long Generalization

Finally, we studied the scenario where labeled data on the target task is difficult and expensive to obtain, but labels on an easier task are more readily available. Specifically, we evaluated the ability of DEAL to generalize from scoring student argument fragments (Franklin et al., 2022) to full student essays (Crossley et al., 2024).

The short dataset consists of individual fragments of student essays corresponding to different components of an argument (e.g. Thesis, Evidence, Rebuttal) and ranges from approximately 10 to 100 words per example. On the other hand, the essay dataset consists of full student essays on a variety of topics (e.g. Mars, Electoral College), and is around 200 to 600 words per essay. Examples are given below:

> Short: "Evidence: It says in paragraph 7, on April 5, 1998, Mars Global Surveyor flew over Cydonia for the first time. Michael Malin took a picture of Mars with his Orbiter Camera, that the face was a natural landform."
>
> Long: I am a scientist at NASA that is discussing the "face" on mars. I will be explaining how the "face" is a land form. By sharing my information about this isue i will tell you just that. First off, how could it be a martions drawing. There is no plant life on mars as of rite now that we know of, which means so far as we know it is not possible for any type of life. That explains...

We transformed the score data into preference data and evaluated both a train on source baseline and DEAL. Results are given in Table 5. Our results indicate that use of labeled argument fragment preference data via DEAL is able to successfully improve performance on the target task of essay scoring, by transferring supervision from the labeled short data to the unlabeled long data. Furthermore, we found that initializing the DEAL model from a snapshot of the train on source baseline at one epoch led to greater performance, even with only one additional epoch of domain adaptation.

While there are significant differences between the two tasks, we believe that there are common, meaningful skills present in both tasks that can be transferred through the use of domain adaptation.

For example, as shown in the example, language conventions (i.e. grammar, punctuation, spelling), basic argument logic, and tone are present in both short and long texts, and we believe that supervision from the short task can effectively improve performance on the long task. We believe that our results provide evidence of the value of DEAL in reducing the need for costly labeling on specialized, difficult tasks, instead allowing humans to express their values on easier tasks and transferring across tasks.

Table 5: Results for short to long generalization with DEAL ($\bar{x} \pm s_{\bar{x}}$ over 3 seeds)

| Method | Pearson's $r$ | **Spearman's $\rho$** |
|---|---|---|
| Train on source | $0.502 \pm 0.013$ | $0.495 \pm 0.011$ |
| DEAL | $0.532 \pm 0.005$ | $0.521 \pm 0.007$ |
| DEAL (init at ep. 1 of train source) | $\mathbf{0.571 \pm 0.006}$ | $\mathbf{0.555 \pm 0.006}$ |
| Train on target | $0.857 \pm 0.003$ | $0.855 \pm 0.003$ |

## 6 CONCLUSION AND FUTURE WORK

In conclusion, we demonstrate the effectiveness of DEAL for aligning LLMs in a variety of scenarios in which data on the target task is not plentiful. For example, we apply our method on transferring supervision between languages and demonstrate that domain adaptation is effectively able to increase performance on a target task without using any labeled data on the target task, instead using labeled data on a similar task. We further apply our method on a few-shot generalization task, where we are given only a low amount of data on a given task but have large quantities of unlabeled data. Using domain adaptation, we effectively map target data points to source data points. This produces a reward model that is tuned to the target evaluation distribution and that effectively uses the labeled few-shot examples combined with the target data. Finally, we apply our method for transferring from an easier, short task to a harder, long task, illustrating the real world potential for DEAL to effectively perform easy-to-hard generalization. This is especially valuable in cases where labels on a hard task may be expensive or difficult to obtain, while labels on an easier task may be more readily available.

In the future, we plan to explore alignment between more drastically different source and target distributions, such as superhuman-level tasks, transferring supervision to LLM-generated samples (including during RLHF training), exploring weak-to-strong generalization, and training a language-fluid LLM. We believe that our work is a significant step forward in effectively aligning large language models in a world with fragmented data.

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

## A EXPERIMENTAL DETAILS

### A.1 GENERAL PARAMETERS

For all experiments, we used base model Gemma-2b (Mesnard et al., 2024) with an additional learned linear head and a learned (low rank adaptation) LoRA (Hu et al., 2021) adapter with rank 64, lora $\alpha$ of 64. We used AdamW (Loshchilov & Hutter, 2017) with learning rate $5e-5$ and no weight decay unless otherwise stated.

For WDGRL, we used $\lambda = 0.01$, $\lambda_{gp} = 1.0$, with 3 critic iterations. The domain adaptation head used a learning rate of 0.0001 with weight decay of 0.001. The domain adaptation head consisted of two MLP layers of width 256 and 128, with GELU (Hendrycks & Gimpel, 2016) activation and no dropout. We used domain adaptation implementations from `https://cpjku.github.io/da/`.

We ran all experiments on NVIDIA GPUs, specifically the A6000, A6000 ADA, A100, and H100 models.

### A.2 ODD ONE OUT EXPERIMENT

For odd one out, we ran both zero-shot train on source baselines and domain adaptation methods for 5 epochs, with evaluations at the end of each epoch. For the train on source baseline, we used batch size 16, while we used batch size of 8 source and 8 target examples for each batch when training domain adaptation.

To generate the odd one out data, we used ChatGPT (OpenAI) and Claude (Anthropic) on Chatbot Arena (Chiang et al., 2024) to create a list of 100 food concepts for each of the following 5 categories: desserts, fruits, sauces, vegetables, and snacks. Examples are given below:

> Desserts: Cake, Pie, Ice Cream, Cookies, Brownies
>
> Fruits: Apple, Banana, Orange, Grape Strawberry
>
> Sauces: Ketchup, Mustard, Mayonnaise, BBQ Sauce, Soy Sauce
>
> Vegetables: Carrot, Potato, Tomato, Onion, Lettuce
>
> Snacks: Potato chips, Pretzels, Popcorn, Cookies, Crackers

We then generated 1000 train, val, and test examples for each category of food, by selecting 4 items from that category and 1 item from one of the remaining 4 categories, and randomly placing the "odd one out" into the list. Our final prompt for the reward model is then:

> Identify the item that does not fit. Only output the name of the item as written and nothing else.
>
> Cake, Pie, Ice Cream, Apple, Cookies
>
> Apple

### A.3 TRANSLATION

For the translation task, we selected three splits (legaladvice, askscience, explainlikeimfive) from the Stanford Human Preference (SHP) dataset (Ethayarajh et al., 2022). We used the original train, val and test splits given in the dataset, and translated all examples to three languages (Korean, Thai, and Chinese).

We used NLLB-200-3.3B (team et al., 2022) translation with temperature 0.0, top_p of 1.0, min_tokens of 0, max_tokens of 1024, and repetition_penalty of 1.15 to reduce repetition in the translations.

We trained both the train on source baseline and the domain adaptation method for 3 epochs, and evaluated every 1000 steps and at the end of each epoch. We used batch size 8 for the train on source baseline and batch size of 4 source examples and 4 target examples for the domain adaptation method.

## A.4 Noise Generalization

For the noise generalization task, we trained both the train on source baseline and the domain adaptation baseline for 3 epochs, with evaluations every 1000 steps and at the end of each epoch. For domain adaptation, we found that using weight decay of 0.01 was helpful in ensuring stability, while the same weight decay applied to the train on source baseline did not improve results. For the train on source baseline, we used batch size 8, while for the domain adaptation method we used a batch size consisting of 4 source examples and 4 target examples.

We used Gemma-2-9b-it (Riviere et al., 2024) to rewrite the Reddit prompts and responses from the Stanford Human Preference dataset, specifically using the prompt "Rewrite this post using highly formal language, using correct grammar, spelling, and punctuation. Expand abbreviations (e.g. aka $\rightarrow$ also known as). Only output the post and nothing else".

## A.5 Few-shot Generalization

For the few-shot generalization task, we trained domain adaptation for 2 epochs, and trained the zero-shot train on source method for 50 epochs to ensure that the same number of passes over the data were allowed during training, with evaluations every 1000 steps and at the end of epochs. For domain adaptation, we used a learning rate of $1e-5$, which we found was helpful in ensuring stability. We used a maximum context length of 768 tokens.

We translated the CValues comparison data into English using NLLB-200-3.3B (same parameters as SHP) and divided the original CValues comparison data into train, validation, and test, while ensuring that the same prompt did not appear in two different splits (we split by prompt). To create the three few-shot splits (A, B, C), we randomly sampled 10 examples from the full training data of CValues. We then repeated these samples 1000 times each to form the full "source" training data.

For the train on source baseline, we used a batch size of 16, while for the domain adaptation method we used a batch size of 8 source examples and 8 target examples. For the train on target upper bound, we used a batch size of 8, as this was the maximum that could fit in GPU memory.

## A.6 Short-to-long Generalization

For short-to-long generalization, we used datasets from Kaggle competitions for argument fragments (Franklin et al., 2022) (short) and full essays (Crossley et al., 2024) (long).

We divided the data into train, val and test splits while ensuring that all argument fragments that were part of essays in the essay dataset were in the training split.

We trained both the train on source baseline, train on target upper-bound, and domain adaptation methods for 2 epochs, with batch size of 32 short examples for the train on source baseline, batch size of 8 long examples for the train on target upper bound, and batch size of 4 source examples and 4 target examples for the domain adaptation methods. For the short examples, we used a maximum context length of 512 tokens, while for the long examples, we used a maximum context length of 1024 tokens.

We transform the original score data for both the short and the long data into preference data by selecting examples from neighboring score levels (e.g. 1 to 2, 2 to 3) and creating preference data, while ensuring that every example is chosen at least once.

