# OpenReview forum: "Aligning Large Language Models with Domain Adaptation"
_ICLR.cc/2025/Conference — ICLR 2025 Conference Withdrawn Submission_

### Official Review · Reviewer_emfu · 2024-10-15

**Soundness:** 3
**Presentation:** 2
**Contribution:** 2
**Rating:** 3
**Confidence:** 5

**Summary:**

This paper introduces Data Efficient Alignment for Language (DEAL), which is a domain adaptation technique employed to align LLMs with human values when labeled target data is scarce or unavailable. DEAL employs Wasserstein Distance Guided Representation Learning to align embedding distributions between source tasks (with labeled data) and target tasks (with unlabeled data), effectively transferring supervision across tasks with similar data distributions. Through a toy experiment and evaluations in four scenarios—cross-lingual transfer, noise resistance, few-shot generalization, and transferring from easy to complex tasks—the authors demonstrate that DEAL enhances LLM performance on target tasks by leveraging source domain data under WDGRL/DEAL.

**Strengths:**

The paper introduces DEAL by applying WDGRL to align LLMs under domain shift in scenarios where labeled target data is scarce.

The experiments introduce and address four specific real-world scenarios of task-specific alignment under domain shift that were previously underexplored.

Exploring domain adaptation for the alignment of LLMs offers a new perspective on transferring supervision across tasks with similar data distributions for alignment.

**Weaknesses:**

The related work section overlooks prior research on domain adaptation in language models and NLP, instead citing some unrelated works from computer vision and robotics. Previous works discussing the limitations of domain adversarial training of language models are missing (e.g. Ruder & Plank 2018,  Karouzos et al. 2021)

In some points in the text (e.g., l 20, l. 182), the term "domain adaptation" is used, implying domain adaptation techniques. Domain adaptation is a term used to describe both a challenge and the methods employed to address that challenge. To avoid confusion, consider specifying 'domain adaptation techniques' or 'domain adaptation methods' when referring to your proposed solutions.

The experimental results present only the proposed method (DEAL), a source-only baseline, and target-only results for one experiment. Comparing your method with other domain adaptation techniques in the context of LLMs alignment would strengthen the evaluation of your work.

The manuscript lacks detailed explanations of methods and experiments, affecting overall clarity and quality.

It's not clear what do you measure and how in the results provided (e.g. Table 2). Is it accuracy of the reward model? Calculated on what test set? Please provide more details on your experimental and evaluation pipeline. What LLM are you using?



References

[Strong Baselines for Neural Semi-Supervised Learning under Domain Shift](https://aclanthology.org/P18-1096) (Ruder & Plank, ACL 2018)

[UDALM: Unsupervised Domain Adaptation through Language Modeling](https://aclanthology.org/2021.naacl-main.203) (Karouzos et al., NAACL 2021)

**Questions:**

1. How DEAL integrates Wasserstein Distance Guided Representation Learning (WDGRL) into the LLM training process?

2. Have you considered comparing DEAL with other established domain adaptation techniques applied to LLM alignment? E.g., pseudo-labeling, domain adversarial training etc.

3. What are the sizes and compositions of your test sets, and how are data splits performed?

4. What training procedures, hyperparameters, and evaluation protocols were followed?

5. Will you be providing code, models, and datasets to facilitate replication of your results?

---

### Official Review · Reviewer_ocM9 · 2024-10-31

**Soundness:** 2
**Presentation:** 2
**Contribution:** 1
**Rating:** 3
**Confidence:** 4

**Summary:**

This paper aims to address the data scarcity issue for preference alignment of LLMs in real-world scenarios. Specifically, the paper identifies four real-world scenarios where there is a lack of target data. These scenarios include transferring supervision across similar tasks, transferring from clean to noisy data, using only a few target examples, and transferring human preferences from easy tasks to more difficult tasks. Meanwhile, the paper proposes a DEAL method to address the data scarcity issues in these identified scenarios. The DEAL method primarily leverages domain adaptation strategies to learn domain-invariant representations for better generalization. The experimental results validate that DEAL effectively aligns capabilities and values of LLMs across related tasks, thereby improving performance on the target task with little to no labeled data.

**Strengths:**

1. Several scenarios introduced in the paper are interesting and worth exploring.
2. The experimental results validate the effectiveness of the methods proposed in this paper.

**Weaknesses:**

1. The paper's motivation lacks clarity and could be further improved. The paper would benefit from providing a more detailed explanation of the close connection between the data scarcity problem in the target task and the four scenarios. The authors should provide a more elaborate explanation of the reasons for utilizing domain adaptation strategies to tackle the data scarcity issue in the target task.
2. The domain adaptation method proposed in the paper appears to be a commonly used approach. It seems that the paper lacks sufficient innovation in its proposed approach.
3. The paper's writing is challenging to comprehend and follow.
4. In the experiments, the paper primarily focused on comparing the proposed method with the "Train on source" and "Train on target" methods, which seems to be insufficient. The comparison should include more domain adaptation and generalization methods.

**Questions:**

Please refer to Weaknesses.

---

### Official Review · Reviewer_M6VP · 2024-11-01

**Soundness:** 2
**Presentation:** 2
**Contribution:** 2
**Rating:** 5
**Confidence:** 3

**Summary:**

This work proposes Data Efficient Alignment for Language (DEAL), using domain adaptation to effectively perform cross-task alignment in scenarios where labeled target data is not available. The authors evaluate DEAL for reward model training on a variety of benchmarks (four real-world LLM alignment scenarios with a lack of target data) and demonstrate that DEAL can meaningfully improve performance on target tasks by utilizing data on related tasks or low amounts of data.

**Strengths:**

1. The proposed method is simple and effective

2. The experiments cover different settings

**Weaknesses:**

1. The authors mention 'reward model' many times. However, it appears unrelated to the actual methods or experiments presented.

2. The writing needs improvement.

3. The rationale behind the method's effectiveness remains unclear. Does it succeed by aligning the representations of source and target tasks?

4. Additional LLMs should be investigated to demonstrate the generalization ability of the proposed method.

**Questions:**

line 201, Tne -> The

---

### Official Review · Reviewer_BLC1 · 2024-11-05

**Soundness:** 2
**Presentation:** 2
**Contribution:** 2
**Rating:** 3
**Confidence:** 4

**Summary:**

This paper addresses four scenarios requiring domain adaptation and introduces an approach called Data Efficient Alignment for Language (DEAL). The experimental results demonstrate that DEAL can enhance performance on target tasks.

**Strengths:**

- The motivation behind selecting the four scenarios is compelling and well-justified.
- The writing of experiments is good.

**Weaknesses:**

- The logical flow and structure of the introduction are lacking. The majority of the introduction is devoted to describing the four scenarios without adequately discussing the method (DEAL) until the last paragraph, and even then, there is no detailed description. As a result, after reading the introduction, I only understand the target tasks without any context on the method or background.
- The related work section fails to clearly differentiate this study from existing domain adaptation research. It is unclear whether the four tasks are newly proposed by the authors or if they follow previous studies.
- Although Figure 1 is visually appealing, it is confusing and difficult to interpret. For instance, in task (1), it appears to be a translation task, but it is unclear which part represents the label and which part represents the similar task. Additionally, the quotation marks in Overleaf should be formatted correctly as ``'' in the caption.
- The description of the method from lines 200 to 204 does not map clearly to Figure 2. Terms like "a main task head" and "a domain critic head" in the text do not correspond directly to "DA Head" and "Reward Head" in Figure 2, making it difficult to understand their equivalence.
- The novelty of the DEAL method appears limited, as the loss function design mainly builds upon the existing WDGRL study. Moreover, the experiments lack comparisons with other existing methods.

**Questions:**

- In line 201, "Tne" should be corrected to "The".
- The paper does not explain why it chose to use translation tasks over more popular domain adaptation tasks. Additionally, there is no justification provided for the lack of comparison with other methods in the experiments.

---

### Note · Authors · 2024-11-21

I have read and agree with the venue's withdrawal policy on behalf of myself and my co-authors.